# Beyond Binary: A Machine Learning Framework for Interpreting Organismal Behavior in Cancer Diagnostics

**DOI:** 10.3390/biomedicines13102409

**Published:** 2025-09-30

**Authors:** Aya Hasan Alshammari, Monther F. Mahdi, Takaaki Hirotsu, Masayo Morishita, Hideyuki Hatakeyama, Eric di Luccio

**Affiliations:** 1Hirotsu Bioscience Inc., New Otani Garden Court 22F, 4-1 Kioi-cho, Chiyoda-ku, Tokyo 102-0094, Japan; hirotsu@hbio.jp (T.H.); m.morishita@hbio.jp (M.M.); h.hatakeyama@hbio.jp (H.H.); 2College of Pharmacy, Mustansiriyah University, Baghdad 10052, Iraq; monther.f71@uomustansiriyah.edu.iq

**Keywords:** organismal biosensing, machine learning, cancer diagnostics, volatile organic compounds (VOCs), behavioral phenotyping, deep learning, precision oncology, *Caenorhabditis elegans*

## Abstract

Organismal biosensing leverages the olfactory acuity of living systems to detect volatile organic compounds (VOCs) associated with cancer, offering a low-cost and non-invasive complement to conventional diagnostics. Early studies demonstrate its feasibility across diverse platforms. In *C. elegans*, chemotaxis assays on urine samples achieved sensitivities of 87–96% and specificities of 90–95% in case–control cohorts (n up to 242), while calcium imaging of AWC neurons distinguished breast cancer urine with ~97% accuracy in a small pilot cohort (n ≈ 40). Trained canines have identified prostate cancer from urine with sensitivities of ~71% and specificities of 70–76% (n ≈ 50), and AI-augmented canine breath platforms have reported accuracies of ~94–95% across ~1400 participants. Insects such as locusts and honeybees enable ultrafast neural decoding of VOCs, achieving 82–100% classification accuracy within 250 ms in pilot studies (n ≈ 20–30). Collectively, these platforms validate the principle that organismal behavior and neural activity encode cancer-related VOC signatures. However, limitations remain, including small cohorts, methodological heterogeneity, and reliance on binary outputs. This review proposes a Dual-Pathway Framework, where Pathway 1 leverages validated indices (e.g., the Chemotaxis Index) for high-throughput screening, and Pathway 2 applies machine learning to high-dimensional behavioral vectors for cancer subtyping, staging, and monitoring. By integrating these approaches, organismal biosensing could evolve from proof-of-concept assays into clinically scalable precision diagnostics.

## 1. Introduction

### 1.1. The Global Cancer Burden and Need for New Diagnostic Approaches

Cancer constitutes a significant and growing challenge to global health. Annually, there are approximately 20 million new cases and 10 million deaths, and these figures are expected to increase markedly in the coming decades [1]. Early and accurate diagnosis is vital to improving patient outcomes. However, current gold-standard diagnostic techniques, despite their clinical effectiveness, have inherent limitations that restrict their widespread global application and impact.

Advanced imaging modalities, including computed tomography (CT), magnetic resonance imaging (MRI), and positron emission tomography (PET), demand substantial capital investment and specialized infrastructure, rendering them inaccessible in many resource-limited settings [2]. Furthermore, the definitive tissue biopsy is an inherently invasive procedure, carrying risks of complications such as infection and bleeding, while also causing considerable patient anxiety [3]. Collectively, these barriers like cost invasiveness, infrastructure, and diagnostic latency constitute a critical gap in modern medicine: the lack of non-invasive, scalable, and cost-effective tools required for effective, widespread population screening.

### 1.2. Organisms as Natural Biosensors

An alternative diagnostic paradigm leverages biology itself. Organismal biosensing exploits the exquisite olfactory capabilities of living organisms as naturally evolved biosensors. The rationale rests on a well-established biochemical principle: cancer alters cellular metabolism in ways that generate disease-specific signatures of volatile organic compounds (VOCs) [4]. These low-molecular-weight metabolites are released systemically and can be detected in accessible biofluids such as breath, urine, and sweat [5]. Across species, evolution has refined sensory systems to detect VOCs at concentrations that surpass the limits of modern analytical instruments [6]. Empirical demonstrations confirm this potential. For example, trained canines have identified lung cancer with high accuracy [7], and the nematode *Caenorhabditis elegans* (*C. elegans*) shows reproducible chemotaxis toward urine from cancer patients [8]. These findings establish organismal biosensing as a credible, non-invasive diagnostic modality.

### 1.3. Problem Statement and the Limitation of Binary Readouts

The development of organismal biosensors as diagnostic tools requires addressing three core challenges: variability, rigor, and scalability. Variability can be reduced by standardizing outputs into discrete measures. In canines, operant conditioning shaped a reproducible “alert” behavior [7], while in nematodes, the Chemotaxis Index (CI) consolidated thousands of trajectories into a population-level scalar [8]. Rigor was established through clinical trial methodologies, including blinding, randomized sample presentation, and appropriate controls [9].

Scalability was achieved through automated imaging and tracking platforms that enabled high-throughput analysis [10]. These innovations produced the first reliable binary outputs, thereby validating organismal biosensing as a diagnostic strategy.

Nevertheless, binary readouts present intrinsic limitations. A simple “cancer present or absent” signal is insufficient for precision oncology, which increasingly requires stratification by cancer subtype, stage, and therapeutic response. Addressing this limitation necessitates new frameworks capable of extracting greater informational depth from organismal behavior.

### 1.4. The Central Thesis in Transition from Signal to Precision

First-generation biosensing demonstrated that cancer-related volatiles can be detected reliably by living organisms [7,8]. However, the informational richness of behavior comprising posture, orientation, and kinematic features remains underutilized. To address this, we introduce the concept of the behavioral vector, B(*t*): a time-indexed, multidimensional representation of organismal behavior, in which each dimension corresponds to a measurable feature (e.g., posture, orientation, trajectory). Temporal dynamics, such as reorientation frequency or rapid turns, add further discriminatory power. Advanced machine learning (ML) models, including convolutional neural networks (CNNs), recurrent neural networks (RNNs), and Transformers, are well-suited to analyze such representations and detect high-resolution diagnostic signatures [11]. As illustrated in Figure 1, we propose a Dual-Pathway Framework for clinical application. Pathway 1 employs validated, low-dimensional indices such as the CI for high-throughput screening, while Pathway 2 applies ML to behavioral vectors for applications in cancer subtyping, staging, and monitoring. The aim of this review is threefold: (i) to outline the biological foundations of organismal biosensing, (ii) to evaluate computational methods for decoding behavior, and (iii) to discuss translational challenges and opportunities for integrating ML into organismal biosensors in oncology.

## 2. Methodology and Scope of the Review

This review employs a narrative synthesis approach to published research on organismal biosensing in cancer detection, with a particular emphasis on integrating machine learning for the interpretation of behavioral signals. Literature was identified through searches of PubMed, Web of Science, and Google Scholar, using combinations of the following keywords: “organismal biosensing,” “cancer detection,” “*Caenorhabditis elegans* chemotaxis,” “canine olfaction cancer,” “insect antennal recording,” “machine learning,” “computational ethology,” and “behavioral signal analysis.” Searches covered peer-reviewed articles published up to August 2025.

Studies were included if they (i) reported experimental or clinical data on non-human organismal biosensing for cancer or disease detection, (ii) described or applied machine learning methods for decoding organismal behavior in a biomedical context, or (iii) presented comparative frameworks, algorithms, or paradigms relevant to high-dimensional behavioral analysis. Exclusion criteria included anecdotal reports, studies without relevance to cancer, and computational models that did not incorporate behavioral or biosensory data.

The scope of this review is intentionally interdisciplinary, spanning oncology, ethology, and computer science. The objective is not to provide an exhaustive catalog of all biosensing studies but rather to (i) compare representative organismal biosensing modalities, (ii) evaluate how machine learning has been or can be applied to these systems, and (iii) propose a unifying Dual-Pathway Framework that links validated binary diagnostic assays with advanced high-dimensional analytics.

## 3. Comparative Analysis of Organismal Biosensors

A robust computational framework must be built upon a critical evaluation of its underlying biological and empirical foundations. Accordingly, this section synthesizes the biochemical evidence for cancer-derived olfactory cues [12] and provides an analytical comparison of the primary animal models used for their detection [7,8]. By dissecting the neurobiological strategies and signal processing principles inherent to each model, we can identify the precise opportunities where computation can unlock a new level of diagnostic insight.

### 3.1. Biochemical Rationale for Cancer’s Volatile Metabolic Signature

The biochemical premise for olfactory biosensing is based on the metabolic reprogramming characteristic of cancer cells. Hallmark shifts toward aerobic glycolysis (the Warburg effect) and altered lipid and amino acid metabolism generate a distinct flux of metabolic byproducts [13]. Many of these are VOCs, which are low-molecular-weight chemicals with high vapor pressure, such as specific aldehydes, alkanes, and benzene derivatives, resulting from processes like lipid peroxidation. Together, these VOCs form a unique, systemic chemical signature of malignancy [12]. Crucially, this “scent of cancer” is not a single biomarker but a complex, high-dimensional signature. It comprises hundreds of chemical species, often at pico- to nanomolar concentrations, posing a sophisticated pattern recognition challenge for which biological olfaction is exceptionally well-suited [14].

### 3.2. Comparative Architectures Across Phyla

Biological systems have evolved a diverse array of solutions to this complex pattern recognition problem. The three models examined here (canines, nematodes, and insects, whose olfactory architectures are illustrated in Figure 2) can be viewed as occupying different points on a spectrum, representing a fundamental trade-off between sensory complexity and experimental tractability. As systematically detailed in Table 1, these biological differences result in fundamentally different data modalities, each with unique advantages and distinct challenges for machine learning.

#### 3.2.1. The Canine Olfactory System and the Challenge of Clinical Translation

The domestic dog (*Canis lupus familiaris*) represents a biological gold standard for olfactory sensitivity. Yet, it also exemplifies the profound challenges of translating a living biosensor into a standardized diagnostic tool. Its power stems from a high-dimensional sensory system employing a combinatorial coding strategy, where distinct odors are represented by unique patterns of receptor activation [22,23]. As summarized in Table 1, this unparalleled sensory complexity is the primary advantage of the canine model, making it an indispensable discovery tool. However, this biological richness comes at a significant cost. Canine assays are inherently low-throughput, labor-intensive, and subject to inter-individual variability, creating fundamental barriers to standardization and scalability. From a data science perspective, this results in a challenging ‘low-N, high-complexity’ problem, making the direct application of data-hungry deep learning models often inappropriate. Consequently, the field must rely on extremely rigorous experimental designs (e.g., triple blinding) and robust statistical methods to ensure the validity of its findings [9]. Thus, the canine model is best positioned not as a scalable screening platform itself, but as a biological benchmark for identifying and validating novel VOC signatures that more scalable technologies can then target.

#### 3.2.2. *C. elegans* as an Optimized, Low-Dimensional Labeled-Line System

In contrast, the nematode *C. elegans* is a low-dimensional system optimized for high-throughput classification. Its entire chemical perception is mediated by only ~32 pairs of chemosensory neurons operating on a “labeled-line” principle, where specific cell types (e.g., AWA, AWC) are tightly coupled to stereotyped attractive or repulsive behaviors [24]. The canonical chemotaxis assay capitalizes on this biological simplicity by employing population averaging as an effective signal-processing strategy [15,25]. By integrating the stochastic movements of thousands of individuals, the Chemotaxis Index leverages the Central Limit Theorem to function as a robust low-pass filter. This filter effectively removes high-frequency noise (uncorrelated individual variations) to preserve the low-frequency signal (the shared directional bias), yielding a highly reproducible readout from a straightforward biological architecture [16].

This approach embodies a deliberate trade-off: it sacrifices sensory complexity and fine-grained behavioral resolution in exchange for scalability and statistical robustness [17]. For machine learning applications, such a simplified, single-feature output is well-suited for training straightforward yet reliable classifiers in binary screening tasks [18]. However, this reductionist framework imposes a significant informational bottleneck. The single CI score discards the majority of the rich kinematic and postural data generated during an assay, limiting its capacity to support more sophisticated diagnostic goals, such as cancer subtyping. Accordingly, while the foundational assay positions *C. elegans* as an exemplary platform for Pathway 1 screening, realizing its full precision potential requires moving beyond the CI toward comprehensive decoding of the behavioral vector.

#### 3.2.3. Insects as a Platform for Decoding the Neural Substrate

Occupying a niche between the behavioral complexity of canines and the high-throughput simplicity of nematodes, insect models offer a powerful platform for directly interrogating neural computation. By performing electrophysiological recordings from ensembles of projection neurons in the antennal lobe (the insect analog of the olfactory bulb), it is possible to decouple sensory perception from motor output [19]. This approach bypasses overt behavior entirely to measure the raw spatiotemporal population vector that constitutes the brain’s primary representation of an odor.

As noted in Table 1, the primary advantage of this model is the direct access it provides direct access to mechanistic insight into the principles of olfactory coding. However, this insight comes at the cost of increased experimental complexity and lower throughput compared to behavioral assays. From a data science perspective, this platform generates a high-dimensional neural time series, a data modality that is an ideal substrate for deep representation learning models (e.g., convolutional neural network–long short-term memory (CNN–LSTM) hybrids) discussed in Section 4. These models can learn to decode diagnostic signatures directly from the patterns of neural activity, providing a powerful method for building and validating computational theories that link chemical inputs to biological signals [20]. Thus, the insect model is best positioned not for mass screening, but as a research platform for developing and testing the advanced algorithms that will be crucial for the entire field.

### 3.3. Portfolio Synthesis and Comparative Insights

Together, these three models illustrate complementary biosensing strategies, each optimized for distinct translational contexts. Canines, with their vast receptor repertoires and combinatorial coding, excel in the discovery of novel cancer-associated VOC signatures [21]. Their sensitivity is unparalleled, but low throughput and variability necessitate strict standardization and blinding protocols [7]. *C. elegans* offers a reproducible, high-throughput platform in which robust population-level chemotaxis indices capture established signatures across large sample sets [26,27]. Insects occupy an intermediate role by offering direct access to neural coding; antennal lobe recordings yield spatiotemporal activity vectors that are ideally suited for mechanistic dissection of olfactory processing and for machine learning applications [27,28].

From a data science perspective, these models produce fundamentally different data modalities: low-N but high-dimensional behavioral classifications from canines; high-N, population-averaged indices from nematodes; and rich neural time-series from insects. This heterogeneity in data modalities is the central computational challenge for the field. It dictates that the choice of a biological platform is simultaneously an a priori commitment to a specific class of machine learning problem, ranging from robust classification on structured data to deep representation learning on complex time series [29,30]. Developing a versatile computational toolkit capable of mastering this full spectrum of challenges is, therefore, the pivotal step toward translating these diverse biological signals into a unified clinical reality, a task we will now explore in detail.

## 4. Machine Learning Frameworks for Decoding Behavioral Signals

The transition from biological observation to clinical diagnosis requires a robust computational layer capable of interpreting complex organismal behavior. While foundational assays validated the principle of organismal biosensing, their reliance on low-dimensional metrics created an information bottleneck, discarding the vast majority of the behavioral data generated. Overcoming this limitation is the central challenge for the field. This section critically analyzes the machine learning paradigms that serve as the interpretive engine for these high-dimensional data streams, evaluating the inherent trade-offs between interpretability, performance, and scalability.

### 4.1. The Conceptual Shift: From Robust Metrics to High-Dimensional Vectors

First-generation methods achieved success by solving a classic signal processing problem: maximizing the signal-to-noise ratio (SNR) for a single, predefined behavioral variable. The *C. elegans* Chemotaxis Index, for instance, is a powerful dimensionality reduction strategy that averages thousands of stochastic individual movements into a single, highly robust scalar value [8,18,31]. This approach deliberately sacrifices granular detail for statistical power and interpretability. Similarly, the discrete “alert” in canine olfaction is an endpoint engineered through operant conditioning to create an unambiguous binary signal [9]. These methods were essential for initial validation but are fundamentally insufficient for the nuanced demands of precision oncology, which require insights into cancer subtype, stage, or therapeutic response.

The contemporary approach re-conceptualizes behavior not as a single variable but as a dynamic trajectory through a high-dimensional state space (a “behavioral vector,” B(*t*)). Formally, this vector represents a multidimensional time series, where each time point contains coordinates describing the organism’s posture and orientation, along with derived kinematic features such as velocity and angular changes. This high-fidelity representation serves as the direct input for the advanced machine learning models discussed in the following sections. This vector encompasses a rich syntax of kinematic motifs (e.g., turns, reversals) and postural dynamics captured by modern pose-estimation tools [12,32]. The analytical goal is no longer to measure a simple directional bias but to decode the information encoded in the structure and sequencing of these behavioral “syllables.” This shift presents a critical trade-off: we gain access to a vastly richer source of diagnostic information at the cost of significantly increased analytical complexity and the risk of overfitting.

### 4.2. Taxonomy of Machine Learning Paradigms

#### 4.2.1. Feature-Based Supervised Learning

Feature-based supervised learning represents a classical, hypothesis-driven approach to machine learning that bridges biological observation with quantitative prediction. This paradigm translates expert knowledge into structured descriptors that algorithms can systematically analyze and process. Its deliberate and transparent nature makes it particularly valuable for early validation and for establishing clinical trust [33].

Unlike data-driven approaches, feature-based methods begin with a biological hypothesis. For example, one may propose that a cancer-associated volatile organic compound profile elicits an aversive response in *C. elegans* [18]. This hypothesis is then operationalized through feature engineering, whereby abstract behaviors are transformed into quantifiable descriptors. Instead of merely observing avoidance, researchers compute measurable parameters such as velocity, pirouette frequency, reversal rates, body curvature, undulation wavelength, and time spent in specific assay zones. These descriptors are assembled into a structured feature matrix, where rows represent trials (e.g., urine samples) and columns correspond to engineered features. It is this matrix, rather than raw trajectories, that is provided to classifiers. Support Vector Machines (SVMs) identify hyperplanes that separate classes within the high-dimensional feature space, while ensemble methods such as Random Forests combine multiple decision trees to improve robustness against noise and reduce overfitting [30].

The principal strength of this paradigm lies in interpretability. Because features are human-defined, model outputs can be traced back to biologically meaningful phenomena [34]. If reversal frequency emerges as a key predictor, it provides evidence that this behavioral element may serve as a biomarker. Post hoc tools, such as SHapley Additive exPlanations (SHAP), further quantify feature contributions, thereby enhancing model transparency [35]. Interpretability is not only valuable for gaining scientific insight but also essential for establishing clinical trust and facilitating regulatory evaluation. Transparent reasoning allows hypotheses to be confirmed or refuted, reassures clinicians that model logic is biologically grounded, and aligns with the emphasis on explainability required by regulatory agencies such as the U.S. Food and Drug Administration [36]. The reliance on expert-defined features, however, also constitutes the main limitation. Models can only detect patterns within the scope of the selected descriptors, making them inherently vulnerable to expert bias. Subtle or composite changes in locomotion may go unnoticed if not explicitly encoded. Furthermore, designing, validating, and maintaining feature sets is labor-intensive and poorly scalable to large or heterogeneous datasets [32].

In summary, feature-based supervised learning provides a rigorous and interpretable framework that remains indispensable for hypothesis testing, biological validation, and the development of trustworthy models from small, well-controlled datasets. At the same time, its dependence on predefined descriptors restricts discovery, leaving more complex or unanticipated patterns to be uncovered by data-driven deep learning approaches.

#### 4.2.2. Representation Learning with Deep Neural Networks: The Pursuit of Performance

While feature-based methods are constrained by pre-existing knowledge, representation learning, powered by deep neural networks enables the automated discovery of predictive features directly from raw data. This discovery-driven paradigm is responsible for state-of-the-art performance in many complex domains and holds immense promise for decoding the high-dimensional behavioral vector [37]. These models typically work as a cohesive pipeline to analyze the different components of behavioral data.

First, CNNs are applied to extract spatial features from individual video frames or postural coordinates. The initial layers of a CNN may learn to detect simple patterns, such as body edges, while deeper layers compose these into complex, stereotyped postural states, all without manual instruction [38].

Next, the sequence of these spatial features is fed into a RNN, often a Long Short-Term Memory (LSTM) unit, which excels at modeling temporal dependencies. The RNN learns the “syntax” of behavior—how postures flow together to form complex movements, such as reversals or dwelling episodes [39]. More advanced Transformer architectures can capture long-range dependencies across an entire behavioral sequence, linking an early sensory cue to a much later action by weighing the importance of all timepoints simultaneously [40]. The primary strength of this end-to-end approach is its ability to reveal unbiased emergent biomarkers, leading to superior predictive performance as datasets grow [37]. However, this power comes with two critical limitations in the biomedical context. First is the “black box” interpretability problem, which hinders clinical trust. This is being actively addressed by Explainable AI (XAI) techniques, such as Grad-CAM, which can generate heatmaps to visualize the trajectory segments most influential to a model’s decision [41]. Second, the data-hungry nature of these models increases the risk of overfitting on small clinical cohorts. This critical challenge is mitigated by strategies such as data augmentation (artificially expanding the dataset) and transfer learning, which adapts models pre-trained on large, general datasets to the specific diagnostic task, significantly improving generalization and reducing data requirements [42].

#### 4.2.3. Unsupervised Learning for Discovering Structure and Generating Hypotheses

The paradigms discussed thus far are supervised, meaning they depend on labeled data (e.g., “cancer” vs. “control”) to guide learning. In contrast, unsupervised learning addresses a deeper question: are there coherent patterns in behavior without any prior labels? This hypothesis-generating paradigm is essential to discovery because it can surface novel behavioral biomarkers that were not anticipated by human intuition.

In practice, unsupervised behavioral discovery typically follows a two-stage workflow. First, dimensionality reduction or embedding techniques transform raw high-dimensional behavioral inputs (for example, time series of posture coordinates) into a latent space. Variational Autoencoders (VAEs) are particularly suitable here: they compress each moment’s posture into a point in a continuous latent space while still allowing for decoding back into the original posture, effectively filtering noise and capturing the essential structure [43]. Second, clustering or density-based methods such as HDBSCAN or visualization-guided clustering on UMAP embeddings are applied to reveal recurring behavioral states. Because these methods do not force clusters into simple geometries, they can uncover arbitrarily shaped clusters corresponding to stereotyped behavioral “syllables,” for instance, fast forward locomotion, local dwelling, or sharp reversals [44]. The output is a data-driven ethogram, which defines behavior as a temporal sequence of discovered states.

The advantage of this approach is that it is unbiased and *de novo*: rather than asking whether worms turn more often, the model may discover that the shift in transition probability between roaming and dwelling is the strongest biomarker, or that rare micro-movements are diagnostic. Such emergent hypotheses can guide downstream mechanistic inquiry.

However, unsupervised discovery comes with a critical caveat: clustering algorithms will always partition data, including random noise, which can lead to over-interpretation of spurious structure. Thus, behavioral clusters must be rigorously validated through multimodal grounding before being accepted as biomarkers. Prominent strategies include chemical grounding (linking a state to specific VOCs measured by GC-MS), neural grounding (correlating a state’s onset with consistent neural activity via calcium imaging), or clinical grounding (showing that the prevalence or duration of a state correlates with cancer subtype, stage, or outcome). A landmark example is the B-SOiD algorithm, which utilized UMAP and HDBSCAN to identify behavioral clusters in mice and cross-validated those clusters against simultaneous neural recordings, confirming that they corresponded to genuine, biologically meaningful behaviors rather than artifacts [45]. Earlier work by Wiltschko et al. further demonstrated that unsupervised embedding and clustering could resolve sub-second behavioral structure in mice, highlighting the utility of such approaches for generating high-resolution ethograms [46].

In summary, unsupervised learning serves as the exploratory engine of the field, as it can generate novel behavioral phenotypes de novo, laying the groundwork for subsequent supervised modeling. But without rigorous multimodal validation, unsupervised findings remain speculative; their real value lies in converting patterns into testable hypotheses that can be linked to biology and eventually deployed in interpretable classifiers for clinical translation.

### 4.3. An Analytical Review of Case Studies

The translation of these distinct machine learning paradigms, from interpretable feature-based models to data-driven deep learning and hypothesis-generating unsupervised methods, into effective diagnostic tools is an emerging process. To ground these theoretical frameworks in practice, the following case studies critically examine how ML has been applied to the unique data modalities of *C. elegans*, canines, and insects, highlighting both current successes and the significant work that remains.

#### 4.3.1. *C. elegans*: A Case Study in Biosensing and ML Integration

The *C. elegans* model exemplifies both the validated success of first-generation biosensing and the immense, yet largely untapped, potential for advanced machine learning integration. Foundational studies have firmly established that nematodes exhibit robust and quantifiable chemotaxis to urine from patients with various cancers, with assays based on the Chemotaxis Index achieving high classification accuracies using classical statistical methods or simple logistic regression [8,18,47].

Furthermore, the dual potential of the platform has been demonstrated by linking these behavioral readouts to underlying neural activity through calcium imaging of olfactory neurons, providing a direct physiological correlate for the diagnostic signal [8,18].

However, to date, the application of advanced machine learning to *C. elegans* cancer diagnostics remains nascent, with most published studies relying on the single-feature CI. While this approach is effective for binary screening, it leaves the rich data of the behavioral vector unanalyzed. The true potential of this platform is therefore best understood by examining the sophisticated ML tools already developed in the adjacent field of *C. elegans* computational ethology. For instance, unsupervised learning methods have successfully decomposed the complete behavioral repertoire of *C. elegans* into a structured set of stereotyped states or “behavioral syllables,” providing a data-driven framework for high-resolution phenotyping. In parallel, deep learning pipelines (e.g., CNN-LSTMs) have been used not only for simple classifications, such as survival state [47], but also to model locomotor trajectories, learning interpretable latent units that correspond to biologically meaningful features, like velocity and curvature, without human supervision.

These enabling technologies, while not yet widely applied to cancer-VOC assays, provide a clear and compelling roadmap. They demonstrate the proven feasibility of using advanced ML to move beyond the informational bottleneck of the CI and decode the entire behavioral vector in this organism [48,49]. This context is critical for interpreting preliminary, non-peer-reviewed data from industry, such as the September 2024 Innovation Update from Hirotsu Bio Science, which announced plans for AI-driven analysis of nematode behavior. While such materials should be viewed with appropriate caution, they signal a clear trajectory toward the integration of the very ML paradigms this review discusses. Thus, *C. elegans* is a key case study not because of widespread current ML application in cancer, but because it represents the most scalable and technically mature platform for deploying these next-generation analytical methods. Full source details, including URLs and supporting slides, are provided in Appendix A.

#### 4.3.2. Canines: A Case Study in ML-Augmented Biosensing

Canine olfaction exhibits remarkable sensitivity to VOCs, but its application in oncology has been limited by variability, methodological heterogeneity, and challenges in scalability. Systematic reviews consistently confirm that trained dogs can achieve high reported accuracies in detecting cancers from breath, urine, or other biological samples; however, they also emphasize substantial differences in blinding, sample preparation, and training protocols that limit reproducibility [7,50,51]. The core challenge lies in converting the subjective cognitive state of the animal (i.e., its decision to “alert”) into an objective, quantitative data stream. The integration of machine learning offers a powerful pathway to bridge this gap.

Rather than analyzing canine behavior directly, the most successful approaches have used the dog as an expert “biological feature selector” to guide the analysis of other data modalities. In a foundational study of prostate cancer, for example, trained dogs were used to identify which urine samples were positive. These canine-flagged samples were then analyzed using gas chromatography-mass spectrometry (GC-MS), and an artificial neural network (ANN) was trained to identify the specific GC-MS peaks corresponding to the dog’s alert. In this workflow, the ML model learns from the chemical data, with the dog’s superior sensitivity used to label the training set [50]. More recently, hybrid platforms have sought to create a standardized signal in parallel with the canine alert. The SpotitEarly system, for instance, reportedly pairs trained dog alerts with algorithmic models that analyze breath VOCs, achieving high reported accuracies across multiple cancers [9]. In such systems, the ML is used to create a reproducible digital signature that can be scaled, using the dog’s performance as a real-time validation benchmark. While the evidence base is still emerging, these examples illustrate ML’s primary role in this context: not as a replacement for the canine’s sensory capacity, but as an essential mechanism to harness and standardize it, converting a variable biological event into a robust dataset suitable for clinical translation [7,51,52].

#### 4.3.3. Insects: A Case Study in Neural Decoding

Insects provide a unique biosensing platform that enables the direct monitoring of volatile detection at the neural circuit level. Farnum et al. demonstrated that population recordings from the locust antennal lobe exposed to oral cancer VOCs achieved classification accuracies ranging from 76% to 100%, with discrimination emerging within approximately 250 ms of stimulus onset, underscoring the ultrafast detection capacity of insect olfactory circuits [20]

Comparable findings have been reported in honeybees, where antennal lobe recordings achieved 82–93% accuracy in discriminating synthetic cancer-related VOC mixtures and lung cancer cell line headspaces, confirming that insect neural circuits can resolve subtle chemical signatures associated with malignancy [53]

Machine learning has played a central role in decoding these high-dimensional neural signals. Support vector machines and random forest classifiers trained on antennal lobe spike trains consistently outperformed traditional rate-based metrics, achieving classification accuracies above 85% in VOC discrimination tasks [24,54]

More recently, Liu et al. combined flexible dual-sided microelectrode arrays with locust antennal lobe circuitry to build a bioelectronic hybrid platform for lung cancer detection, reporting 100% classification success for known VOCs and ~85% accuracy for human lung cancer cell line discrimination [54].

These studies highlight insects as a particularly promising model for bio-hybrid sensor development, in which biological neural signals are directly interfaced with computational classifiers to enable rapid and high-accuracy detection.

Nonetheless, limitations remain. Most experimental paradigms rely on invasive electrophysiology, which complicates long-term or portable monitoring. Consequently, the majority of assays have been restricted to culture-derived VOCs or synthetic biomarker mixtures, rather than clinical breath or urine samples. The translational challenge, therefore, lies in bridging proof-of-concept laboratory studies with clinically validated bio-hybrid platforms. Table 2 provides a comparative synthesis of machine learning paradigms applied across organismal biosensors, summarizing representative cancer types, reported metrics, and the respective strengths and limitations of each platform.

### 4.4. Methodological Rigor and the Hurdles to Clinical Translation

Beyond conceptual frameworks, the successful deployment of any ML model hinges on addressing a set of critical implementation challenges that directly impact its reliability, reproducibility, and ultimate clinical utility. These are not minor technical details; they are fundamental hurdles that must be overcome for organismal biosensing to mature into a trustworthy diagnostic modality.

#### 4.4.1. The Pervasive Challenge of Class Imbalance

In realistic cancer screening cohorts, the number of “healthy” or control samples vastly exceeds the number of cancer-positive cases, reflecting the natural prevalence of the disease. This severe class imbalance represents one of the most critical pitfalls in medical machine learning. A naïve classifier can appear to achieve >99% accuracy simply by predicting the majority class (i.e., “no cancer”), yet such a model is statistically misleading and clinically useless [55].

The clinical consequences of ignoring class imbalance are profound. A model with low sensitivity (recall) will fail to identify true cancer cases, thereby delaying life-saving interventions. In contrast, the consequence of false positives, although associated with anxiety and additional diagnostic testing, is generally less severe. This asymmetry establishes a clear clinical mandate: screening models must be explicitly optimized to maximize sensitivity, even at the expense of specificity [55,56].

The problem is compounded in organismal biosensing assays, where the minority (cancer-positive) class is not only small but also highly heterogeneous. Distinct cancer subtypes, stages, and patient-specific metabolic states may elicit subtly different behavioral or neural responses in the biosensor, producing a complex multi-modal minority class that is inherently difficult to model from limited data [55,56].

A range of algorithmic approaches has been proposed to mitigate imbalance. Resampling techniques, such as the Synthetic Minority Over-sampling Technique (SMOTE), are widely used to balance class distributions numerically [57]. However, in high-dimensional behavioral or neural trajectory data, synthetic oversampling risks generating biologically implausible patterns, potentially teaching the model to recognize artifacts rather than genuine disease signals [58]. A more clinically aligned approach is cost-sensitive learning (CSL), which directly encodes the relative importance of different types of classification errors into the model’s loss function. By assigning a much higher penalty to false negatives than to false positives, CSL forces the model to learn the discriminative features of the rare cancer-positive class, regardless of how subtle and heterogeneous they may be [59]. Recent reviews emphasize that CSL consistently outperforms naïve resampling methods in healthcare contexts and represents an essential strategy for developing clinically responsible screening models [55].

In summary, class imbalance is a pervasive methodological barrier that undermines both statistical validity and clinical utility. Effective solutions will require a combination of principled algorithmic approaches, rigorous validation, and careful evaluation of sensitivity-specificity trade-offs to ensure that biosensor-based screening tools achieve both robustness and clinical relevance.

#### 4.4.2. Overfitting and the Challenge of Biological Variability

In organismal biosensing, the risk of overfitting, where a model learns spurious details of the training set rather than generalizable biological principles, is particularly high [60]. Such models may appear accurate in internal testing but, similar to the classical “Clever Hans” analogy, succeed only by exploiting irrelevant cues instead of detecting the true biological signal [60]. Overfitting arises from two principal sources: spurious correlations within small and heterogeneous patient cohorts, and confounding variables introduced by the biosensing platform itself (e.g., batch effects). For example, classifiers may inadvertently learn to associate diagnostic labels with technical artifacts (e.g., batch effects, culture differences, lighting, or instrument drift) rather than true biological signals [61,62].

When trained on such environment-specific confounders, models almost invariably fail under domain shift, where data are collected at different sites, times, or using alternative equipment [63]. For this reason, rigorous external validation, in which models are tested on independent datasets generated in distinct environments, remains the gold standard for evaluating robustness. Models validated only by internal cross-validation cannot be regarded as reliable, and their reported performance must be interpreted with caution. Several methodological strategies mitigate overfitting [64]. Regularization techniques such as Dropout and L_2_ weight decay constrain model complexity and reduce reliance on any single feature, encouraging the extraction of more generalizable representations [65].

Data augmentation, widely used in biomedical imaging and behavioral datasets, artificially expands training sets by applying controlled perturbations (for example, rotations, flips, brightness changes, or noise), thereby teaching models to recognize biological signals independent of nuisance variability [66].

Finally, transfer learning has become a critical strategy in small-data biomedical contexts. Models pretrained on large and diverse datasets capture general spatiotemporal features that can then be fine-tuned on smaller biosensing cohorts, improving generalization and reducing the likelihood of overfitting [67,68].

In summary, overfitting and biological variability pose significant challenges to the development of reliable biosensor-based diagnostic models. Addressing these challenges requires not only technical safeguards such as regularization, augmentation, and transfer learning but also stringent external validation to ensure that models capture genuine biological signals rather than artifacts of a particular dataset or laboratory environment.

#### 4.4.3. Evaluation Metrics: Beyond Simple Accuracy to Clinical Utility

In cancer screening, reliance on “accuracy” as the principal evaluation metric is not only inadequate but also misleading. In an imbalanced dataset, a model may appear to achieve 99% accuracy by consistently predicting the majority class (i.e., “no cancer”) for every case, yet fail to identify any true positives. Such a model has no diagnostic value, underscoring the necessity of evaluation frameworks that reflect the clinical priorities of screening [69].

The performance of a screening tool is more meaningfully assessed through sensitivity and precision, which capture two complementary aspects of clinical performance. Sensitivity (also known as recall) measures the proportion of true cancer cases that are correctly identified as such. It is crucial in screening, as false negatives represent missed opportunities for timely intervention, which can have potentially fatal consequences [70]. Precision measures the proportion of predicted positives that are correct, reflecting the burden of false positives. While reduced precision does not endanger patients directly, it generates psychological distress and incurs financial and clinical costs through unnecessary follow-up procedures [71].

These two dimensions are inherently in tension with each other. A model optimized for sensitivity will identify subtle anomalies and thereby generate more false positives, whereas a model tuned for precision will adopt a conservative threshold, inevitably missing indolent or early-stage cases. To evaluate this trade-off comprehensively, the Precision–Recall Curve (PRC) provides a threshold-independent summary, with superior models characterized by curves shifted upward and to the right. The Area Under the PRC (AU-PRC) condenses this curve into a single value that emphasizes performance on the minority class. By contrast, the more familiar Area Under the ROC Curve (AUC-ROC) is often misleading in imbalanced datasets because it is disproportionately influenced by the true negative rate, reflecting the relative ease of identifying the majority control class rather than the challenge of detecting rare positives [69,72].

For these reasons, AU-PRC, together with transparent reporting of the full confusion matrix and the use of robust validation strategies such as stratified k-fold cross-validation, should be regarded as the minimum methodological standard in this field. Only by adopting metrics that align with the clinical priorities of sensitivity and precision can biosensor-based models be evaluated with both statistical rigor and translational relevance.

#### 4.4.4. Computational Feasibility and the Path to Point-of-Care Deployment

Although GPU-accelerated pipelines enable the training of complex models in research environments, the ultimate clinical value of organismal biosensing depends on its deployment in accessible and often resource-limited settings. Computational feasibility, therefore, shifts from a technical detail to a fundamental design requirement, particularly in the context of Pathway 1 of the proposed Dual-Pathway Framework. The promise of a low-cost, scalable screening tool for global health can only be realized if analytical models are capable of running efficiently on local, non-specialized hardware [73].

The engineering challenge lies in translating large, computationally intensive models developed in high-performance computing environments into lightweight implementations suitable for point-of-care deployment. This process, commonly referred to as model compression or optimization, is a critical step for clinical translation. Several strategies are central to this effort. Model pruning systematically removes redundant connections or neurons from a trained network, analogous to trimming branches from a tree, thereby reducing both memory footprint and inference time while preserving predictive performance [74]. Quantization reduces the numerical precision of model parameters, for example, by converting 32-bit floating-point weights into 8-bit integers, which substantially decreases storage requirements and accelerates inference on CPUs or low-power hardware without significant loss of accuracy [75]. Knowledge distillation adopts a teacher–student paradigm, where a large, accurate model transfers its predictive capacity to a smaller and more efficient model, enabling the latter to approach the performance of the former with a fraction of the computational cost [76].

The successful application of these strategies enables real-time inference on widely available edge devices such as single-board computers (e.g., Raspberry Pi) and even consumer smartphones. Recent demonstrations of under 200 ms inference times for real-time insect neural decoding confirm that high-performance analysis is achievable outside of specialized computing facilities [20]. Prioritizing the development of such optimized architectures is therefore essential to bridge the gap between research prototypes and clinically viable, globally scalable biosensing platforms.

## 5. A Proposed Dual-Pathway Framework for Clinical Implementation

The preceding analysis reveals a fundamental tension in organismal biosensing: the need for both simple, interpretable assays for accessible screening and complex, high-performance models for precision diagnostics. To address this challenge, we propose a Dual-Pathway Framework that strikes a balance between accessibility and precision by dividing the workflow into two complementary tiers. Pathway 1 utilizes validated, biologically driven assays, such as chemotaxis indices or trained behavioral alerts, for accessible and low-resource screening. Pathway 2 builds on these foundations by applying advanced machine learning to extract high-resolution behavioral features, enabling refined diagnostic applications in precision oncology.

This tiered design reflects established precedents in cancer diagnostics, such as reflex triage in HPV testing [77] and the REASSURED criteria for global health diagnostics [78]. By integrating the scalability of organismal biosensors with the analytical depth of modern computation, the Dual-Pathway Framework provides a pragmatic roadmap for clinical deployment. As illustrated in Figure 3, biosensing is positioned not as an alternative to standard diagnostics but as a complementary platform that can expand screening reach and enhance diagnostic resolution across diverse healthcare settings.

### 5.1. Pathway 1: Biologically Driven Screening for Scalability and Access

Pathway 1 is designed for accessible, high-throughput screening in resource-constrained environments. This pathway leverages the foundational, low-dimensional signals discussed earlier, such as the *C. elegans* Chemotaxis Index, which distills thousands of complex trajectories into a single, robust, and easily interpretable score [8,77].

The core of this pathway is a deliberate trade-off: it sacrifices granular diagnostic information (e.g., cancer subtype) for maximum simplicity, low cost, and scalability. Assays like the nematode urine screen require minimal infrastructure and can be deployed in community health centers, directly addressing the global need for accessible diagnostics [10]. The core design of this pathway involves a deliberate trade-off, sacrificing diagnostic granularity (e.g., cancer subtype) for maximum simplicity, low cost, and scalability. While this design makes assays like the nematode urine screen highly deployable [9], it also introduces profound implementation challenges. Reducing a complex behavior to a single score introduces the critical challenge of defining an appropriate diagnostic threshold. This cutoff is not absolute and must be carefully calibrated; a threshold set to maximize sensitivity will, in a low-prevalence population, result in a low Positive Predictive Value (PPV), leading to an overload of the healthcare system with false positives that overwhelm follow-up capacity and cause significant patient anxiety [79,80].

Furthermore, the very simplicity that makes the score accessible also creates a significant risk of misinterpretation, where a “high-risk” result can be mistaken for a definitive diagnosis, leading to significant patient anxiety from potentially false-positive results [81]. Therefore, the successful implementation of Pathway 1 is not merely a technical challenge but a socio-technical one, whose value is entirely contingent on the existence of a robust infrastructure for confirmatory testing and treatment, a core principle of any effective screening program [82]. This necessitates the development of clear communication protocols to manage the clinical and psychological impact of its results, ensuring that patients and providers alike understand the preliminary nature of a screening test.

### 5.2. Pathway 2: Machine Learning-Augmented Diagnostics for Precision and Depth

Pathway 2 is designed to maximize informational yield, providing the high-resolution diagnostics required for precision oncology. This pathway leverages the full range of machine learning paradigms discussed in Section 4 to analyze complex, high-dimensional behavioral vectors. By employing architectures ranging from CNNs to Transformers, it is possible to decode subtle kinematic and postural signatures. In doing so, this extends beyond simple risk stratification to address nuanced clinical questions such as cancer subtype discrimination, staging, and monitoring of therapeutic response [83]. This distinction represents the difference between merely flagging a sample as “high risk” and generating actionable clinical intelligence, for example, identifying an aggressive triple-negative breast cancer that necessitates immediate, targeted therapy. However, the realization of this advanced diagnostic layer depends on overcoming a series of interconnected translational challenges [84].

The most formidable barrier is crossing the “validation chasm” that separates a high-performing algorithm from a clinically approved diagnostic tool. Because such models would be classified as Software as a Medical Device (SaMD), they are subject to rigorous regulatory frameworks that demand empirical evidence of analytical validity, clinical validity, and ultimately, clinical utility [85]. Demonstrating these requirements necessitates large-scale, multi-center prospective trials with tissue-based confirmation and long-term follow-up, representing a substantial investment of time and resources. Meeting these stringent validation demands also raises critical questions of data governance and equity [86]. The diverse and expansive datasets required for model development introduce significant privacy concerns, underscoring the need for privacy-preserving strategies, such as federated learning, to manage sensitive patient information across institutions securely [87]. Equally important, failure to ensure representativeness risks exacerbating health disparities by producing models that generalize poorly to underrepresented populations, thereby deepening existing inequities in care [88].

Finally, even models that are validated and ethically developed face a persistent barrier to clinical adoption: the issue of interpretability. Clinicians must be able to trust and act upon high-stakes predictions from complex models, which requires access to a clear rationale. Pathway 2 must therefore be engineered with transparency by incorporating Explainable AI methods. Such approaches not only facilitate error analysis and strengthen clinical confidence but are increasingly regarded as essential prerequisites for regulatory approval [89].

### 5.3. A Tiered Diagnostic Workflow: Integrating Both Pathways

The true strength of the Dual-Pathway Framework lies in integrating both pathways into a tiered, “reflex” diagnostic workflow. This design is not novel, but mirrors established and successful screening paradigms, such as reflex triage in human papillomavirus (HPV) testing, which has been shown to improve both clinical efficiency and cost-effectiveness [90].

The underlying health-economic rationale is to apply a low-cost, highly scalable tool (Pathway 1) for broad population screening, while reserving more complex and resource-intensive diagnostics (Pathway 2 and subsequent clinical investigations) for the smaller subset of individuals identified as high risk. Such a strategy aligns with the central objectives of evidence-based cancer screening programs, namely optimizing resource allocation and minimizing unnecessary interventions [91].

Population screening and cancer subtyping. As a primary screening tool, a positive result from a low-cost *C. elegans* urine test (Pathway 1) would trigger reflexive high-resolution analysis of the raw behavioral data via a Pathway 2 model. This layered approach moves beyond a binary result to generate clinically actionable intelligence. In breast cancer, for instance, such reflexive analysis could help distinguish an aggressive triple-negative subtype requiring urgent biopsy from a less aggressive luminal A subtype, thereby directly shaping the next step in patient management [92].

Non-invasive Recurrence Monitoring. For patients in remission, ongoing surveillance is essential but often depends on costly or invasive modalities such as intermittent colonoscopy in colorectal cancer. A low-cost, non-invasive Pathway 1 test administered at regular intervals (e.g., every few months) could provide a more vigilant and patient-friendly surveillance option. A transition from a “no-risk” to a “high-risk” signal would act as an early warning of recurrence, prompting confirmatory diagnostic testing long before clinical symptoms arise. This approach addresses a significant unmet need in survivorship care, where earlier detection of recurrence can significantly improve outcomes [93].

Longitudinal Monitoring of Therapeutic Response. A forward-looking application of the reflex framework is dynamic tracking of treatment efficacy. Cancer therapies, including chemotherapy and immunotherapy, vary widely in their effectiveness, and clinical evidence of benefit often takes months to establish. By serially analyzing the behavioral vector with Pathway 2, subtle changes in a patient’s volatile organic compound (VOC) profile may be detected over the course of therapy. This could provide an early, non-invasive indication of treatment response or resistance, enabling clinicians to make more timely and informed adjustments to therapeutic regimens [94].

By functioning as an intelligent front end to the existing diagnostic ecosystem, this tiered reflex framework does not aim to replace gold-standard modalities such as imaging or biopsy. Instead, it augments clinical decision-making by ensuring that resource-intensive diagnostic tools are deployed more efficiently and precisely in the patients who stand to benefit most.

## 6. Discussion

Translating the proposed dual-pathway framework from a conceptual model into a reliable clinical tool presents significant but addressable challenges. Realizing this vision requires robust, reproducible processes that can perform consistently across diverse clinical and laboratory settings. This discussion synthesizes the key translational hurdles from data standardization to regulatory approval and outlines the next frontier of innovation for the field

### 6.1. A Foundational Challenge in Building a Standardized Data Ecosystem

The advanced machine learning frameworks described earlier offer tremendous potential; however, the quality and consistency of the underlying data fundamentally constrain their impact. The single most significant scientific barrier to clinical translation of organismal biosensing is the absence of standardized, reproducible behavioral datasets. Behavior is inherently dynamic and high-dimensional, and unlike static modalities such as genomics, it is exquisitely sensitive to subtle environmental or procedural differences, such as ambient temperature, agar concentration, lighting conditions, or handling protocols that vary across laboratories [95,96]. This inter-laboratory variability is a major contributor to domain shift, wherein models trained in one laboratory fail to generalize in another setting [97,98]

Mitigating this challenge requires a community-wide commitment to consensus protocols and standardized platforms. Examples include microfluidic arenas for precisely controlled VOC delivery, unified handling and maintenance routines for organisms, and strict environmental controls for behavioral assays. These standards form the necessary baseline for reproducible data acquisition [99,100]

However, protocol standardization alone is insufficient. The field’s maturation into a true diagnostic science depends on creating a large-scale, multi-species Behavioral Diagnostics Repository. Analogous to foundational datasets like ImageNet, such a repository would compile annotated behavioral and neural recordings linked to validated clinical metadata, enabling cross-species transfer learning and robust model training. Initiatives in other AI domains emphasize that reproducibility and generalizability are strengthened when models are trained and benchmarked on shared, large-scale datasets [101]

Building such a repository entails overcoming both technical and cultural obstacles: defining a common behavioral ontology for consistent annotation (so that data collected by different groups is interoperable) and surmounting academic data silos that hinder open sharing. Ultimately, the establishment of this shared data ecosystem is the critical inflection point for organismal biosensing. It will enable the transition from isolated, small-scale laboratory studies to a cumulative, data-driven discipline capable of producing globally validated diagnostics. For a practical summary of minimal specifications and standardization guidelines, see Appendix A.

### 6.2. Navigating the Regulatory and Ethical Landscape

Beyond scientific challenges, the clinical translation of organismal biosensing depends on successfully navigating a tightly interwoven framework of regulatory compliance and ethical oversight, which together underpin clinical and public trust. The hybrid nature of these diagnostics, which combine living biology with AI-driven Software as a Medical Device (SaMD), raises novel regulatory questions that demand a convergent approach. The biological component may be subject to manufacturing and biosafety regulations (e.g., GMP, organism stability), whereas the computational component demands rigorous validation, algorithm auditing, and post-market surveillance FDA, 2025 [102].

A coherent regulatory pathway must integrate both streams: guaranteeing the reliability and stability of the biological sensor under controlled conditions, and ensuring that the AI module complies with established standards for transparency, risk management, and lifecycle monitoring. Challenges such as defining “locked” versus “adaptive” algorithms, setting change control plans, and managing real-world learning updates are central to the safe deployment of AI in clinical practice [103]

Collaborative engagement between developers, clinicians, and regulators will be essential for establishing these integrated validation pathways.

Ethical oversight is inseparable from regulatory validation and rests on three interconnected pillars. The first is animal welfare: the use of sentient organisms, particularly canines, must comply with international principles and oversight by Institutional Animal Care and Use Committees (IACUC), recognizing that reproducibility is itself an ethical concern in animal studies [104]

The second is data privacy and governance: protection of patient data is mandated under legal frameworks such as HIPAA and GDPR and is increasingly addressed through privacy-preserving machine learning strategies, including federated learning, which enable secure multi-institutional collaboration without compromising data sovereignty [105].

The third is public and clinical acceptance: novel diagnostics involving living organisms may encounter cultural skepticism. Building acceptance requires transparent communication, participatory public engagement, and education strategies to ensure informed trust and facilitate recruitment into validation trials [106].

In sum, the successful translation of organismal biosensing requires a holistic strategy in which technical validation, regulatory compliance, and ethical stewardship are pursued in parallel as an integrated framework. Only through such convergence can these novel diagnostics progress from promising laboratory innovations to clinically trusted and socially legitimate tools.

### 6.3. The Next Frontier of Organismal Diagnostics

Successfully navigating the challenges of standardization and regulation will unlock a transformative future for organismal diagnostics. The next frontier will be defined not by incremental improvements, but by synergistic advances that reposition these platforms from standalone tests into integrated components of proactive, personalized health monitoring.

One immediate frontier is the integration of multimodal data. A behavioral risk score gains significantly greater power when combined with complementary data streams, such as genomic risk profiles, metabolomics, and real-time physiological inputs from wearable sensors. Multimodal AI is already showing promise in medical diagnostics by combining heterogeneous modalities (images, signals, clinical records) to improve robustness and interpretability [107,108]. For example, longitudinal integration of non-invasive data has been proposed to enhance survival predictions in cancer immunotherapy by fusing blood, imaging, and clinical metadata [109].

As ensemble models become increasingly complex, their clinical adoption will depend on their explainability. Clinicians must be able to understand how a model balances behavioral markers against genetic or metabolic features. Models that provide transparent, interpretable outputs are more trustworthy and audit-friendly, an essential prerequisite as complexity increases.

A complementary, long-term frontier lies in synthetic biology, which aims to create programmable diagnostics. Rather than simply observing natural behavior, organisms could be engineered with bespoke sensory capabilities. Tools such as CRISPR-Cas9 and synthetic genetic circuits enable the programming of organisms to express novel olfactory receptors or sensors coupled with fluorescent, colorimetric, or other quantifiable reporters [110,111,112]. For instance, programmable cell-based biosensors have been proposed in the synthetic biology literature for environmental and medical sensing contexts [113]. While these “living diagnostics” face significant regulatory and biosafety challenges, they hold promise as highly specific, low-cost sensors customizable to a broad array of disease signatures.

Ultimately, these frontiers converge toward a paradigm shift toward integrated health monitoring. In this vision, continuous streams of behavioral, molecular, and physiological data, combined with programmable biosensors, shift oncology from episodic screening to proactive surveillance. Changes in a patient’s “volatome,” continuously monitored through these tools, could trigger alerts at the earliest stages of disease, enabling timely interventions when treatment is most effective.

## 7. Conclusions

This review has synthesized interdisciplinary evidence to establish a framework for interpreting complex organismal behaviors in cancer diagnostics. Our central finding is that machine learning provides an essential interpretive engine to decode the high-dimensional behavioral signals elicited by cancer-associated VOCs. While the field is clearly moving beyond simple binary readouts, its clinical translation remains constrained by methodological heterogeneity, a lack of large-scale standardized datasets, and the challenge of model interpretability.

To address these challenges, we have proposed a Dual-Pathway Framework as a pragmatic roadmap for clinical implementation. This model resolves the inherent tension between access and precision by synergizing validated, low-resource biological assays for broad, population-level screening with AI-augmented analytics for high-resolution, precision diagnostics.

The maturation of this field from proof of concept to clinical reality is therefore contingent upon a concerted, future-focused effort. Researchers must prioritize the development of (1) robust, multi-species behavioral databases to enable generalizable models, (2) explainable AI techniques to build clinical trust, and (3) large-scale, prospective validation studies to prove clinical utility. By systematically addressing these challenges, the convergence of organismal biosensing and machine learning is poised to deliver the next generation of non-invasive, scalable, and clinically impactful tools for precision oncology.

## Figures and Tables

**Figure 1 biomedicines-13-02409-f001:**
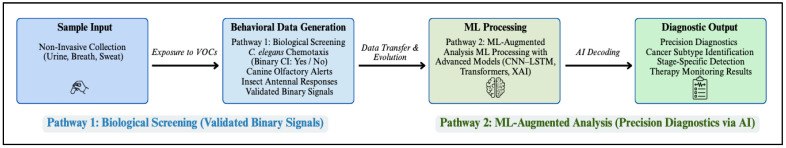
Transition from binary screening to AI-augmented diagnostics. Biological samples containing cancer-associated volatile organic compounds (VOCs)—for example, urine, breath, or sweat—can elicit measurable behavioral responses in living biosensors. Pathway 1 (validated screening) relies on established low-dimensional metrics, such as the *C. elegans* Chemotaxis Index, to generate a binary output optimized for high-throughput population screening [8,10]. In contrast, Pathway 2 (AI-augmented diagnostics) treats raw behavioral data as a high-dimensional behavioral vector, B(*t*), analyzed with advanced machine learning models (e.g., convolutional neural networks [CNNs], recurrent neural networks [RNNs], Transformers). This richer representation enables not only cancer detection but also finer-grained tasks such as subtype discrimination, staging, and therapeutic monitoring [11,12].

**Figure 2 biomedicines-13-02409-f002:**
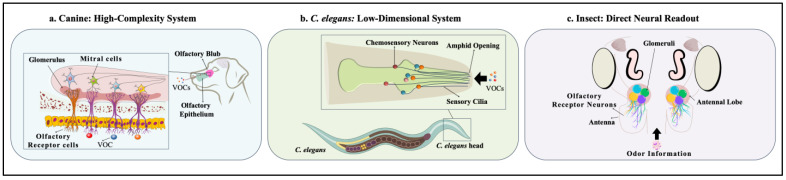
Three representative strategies are shown: Three distinct systems illustrate complementary strategies for VOC detection and neural decoding: (**a**) Canine olfaction (*Canis lupus familiaris*): A high-dimensional combinatorial code employing hundreds of olfactory receptor neuron (ORN) types projecting to glomeruli in the olfactory bulb [15,16,17,18]. *Color key:* ORNs are shown in different colors to indicate distinct receptor types; VOCs are represented by multi-colored spheres (red, yellow, orange); glomeruli are shaded green; mitral cells are light blue/purple. (**b**) Nematode olfaction (*C. elegans*): A low-dimensional labeled-line system in which ~32 pairs of chemosensory neurons (e.g., AWA, AWC; depicted in various colors) drive stereotyped chemotactic behaviors [15,16,17,18]. *Color key:* Chemosensory neurons and their sensory cilia are shown in distinct colors (green, blue, red, orange) to denote different neuron classes; stimuli odorants entering the amphid opening are represented by matching colored spheres. (**c**) Insect olfaction (generalized model): A direct neural readout strategy where olfactory receptor neurons (ORNs) project to the antennal lobe, enabling rapid decoding of VOC-induced neural population activity [19,20,21]. *Color key:* ORNs are color-coded (purple, yellow, blue, green) according to receptor specificity, projecting to corresponding color-coded glomeruli in the antennal lobe. Figure created by Aya Hasan Alshammari (author) using Microsoft PowerPoint.

**Figure 3 biomedicines-13-02409-f003:**
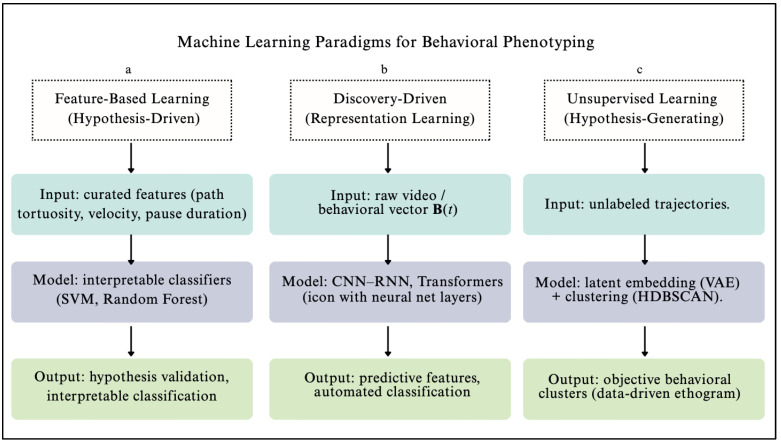
Machine learning paradigms for decoding organismal behavior. Pathway 2 of the dual-pathway framework integrates three paradigms: (**a**) feature-based learning (hypothesis-driven) applies biologist-defined features with interpretable models such as SVMs and Random Forests [30,34,35]; (**b**) representation learning (discovery-driven) uses deep neural networks, including CNNs, RNNs, and Transformers, to automate feature discovery [37,38,39,40]; and (**c**) unsupervised learning (hypothesis-generating) employs latent embeddings (e.g., VAEs) and clustering algorithms such as HDBSCAN to reveal novel behavioral motifs and construct data-driven ethograms [43,44,45,46]. Notes: CNN, convolutional neural network; RNN, recurrent neural network; SVM, support vector machine; XAI, explainable artificial intelligence; VAE, variational autoencoder; HDBSCAN, hierarchical density-based spatial clustering of applications with noise.

**Table 1 biomedicines-13-02409-t001:** Comparative features of organismal biosensing platforms.

Organism	Olfactory Strategy	Sensory Complexity	Primary Data Modality	Experimental Throughput	Scalability	Key Advantage	Key Limitation	Primary ML Challenge	References
Nematodes (*C. elegans*)	Low-dimensional labeled-line system	Very low (~32 chemosensory neuron pairs)	High-N, scalar population index (Chemotaxis Index)	Very high (hundreds of samples/days with automation)	High (amenable to microfluidics and automation)	High-throughput screening for validated signals	Informational bottleneck of the single CI	Extracting nuanced signals beyond the CI	[8,10,17,18,19,20,21,22]

Canines (*Canis lupus familiaris*)	High-dimensional combinatorial code	Very high (>800 olfactory receptor genes)	Low-N, discrete behavioral classification (alert/no alert)	Very low (hours to days per cohort)	Low (requires individualized training)	De novo discovery of novel VOC signatures	Lack of scalability and inter-animal variability	Few-shot learning, mitigating inter-animal variability	[6,7,8,9]

Insects (locusts, honeybees)	Intermediate-dimensional combinatorial code	Medium (~60 glomeruli in antennal lobe)	High-dimensional, spatiotemporal neural time series from antennal lobe recordings	Medium (minutes per sample; constrained by electrophysiology)	Medium (requires specialized neural recording infrastructure)	Mechanistic insight into neural coding and rapid classification	Experimental complexity and invasiveness of neural recordings	Decoding complex, high-dimensional neural codes	[19,20]

Notes: OR = olfactory receptor; CI = Chemotaxis Index; VOC = volatile organic compound.

**Table 2 biomedicines-13-02409-t002:** Summary of machine learning applications in organismal biosensing for cancer diagnostics.

Organism	ML Method and Paradigm	Example Cancer Type	Key Reported Metrics	Key Advantages	Key Limitations and Challenges	Ref
*C. elegans*	Chemotaxis index analyzed with logistic regression; calcium imaging classified with SVMs; enabling ML pipelines (CNNs, LSTMs, RNNs) developed for tracking and phenotyping (not yet applied directly to cancer urine)	Breast (urine), prostate (urine), multi-cancer (urine)	Chemotaxis assay: sensitivity 95.8%, specificity 95.0% (n = 242; threshold analysis); Combined dilution assay: sensitivity 87.5% (n = 78; blinded, case–control)	Non-invasive urine workflow; mechanistic link to olfactory neurons; scalable population screening; ML expands quantification and mechanistic inference	Small cohorts; prospective validation needed; ML-driven behavioral phenotyping is not yet clinically validated for cancer screening	[8,22,47,48,49]

Canines	ANN trained with canine-labeled GC–MS; AI-assisted ensemble models for breath samples	Prostate (urine), multi-cancer (breath), osteosarcoma (saliva)	Prostate urine pilot: sensitivity 71%, specificity 70–76% (n ≈ 50; case–control). AI–canine breath platform (SpotitEarly): accuracy 94–95% (n ≈ 1400; multi-cancer). Osteosarcoma: cell line blind trials, sensitivity 97.7%, specificity 98.6% (n ≈ 200); saliva pilot (2 patients) showed near-perfect discrimination.	High early-stage sensitivity; non-invasive (urine, breath, saliva); AI improves standardization and multi-cancer capability	Performance varies across cancer types, with limitations in reproducibility, dependence on intensive training, and ongoing challenges in scalability and logistics.	[7,9,50,51,52]

Insects	Antennal-lobe neural recordings with supervised decoding	Oral cancer (locust), lung cancer, VOCs and cell lines (honeybee)	Locust antennal lobe: classification accuracy 100% for unknown VOCs, 86% for concentration variation, 85% for human lung cancer cell lines (n ≈ 20–30; cross-validation). Honeybee antennal lobe: 88% (single VOC biomarkers), 93–100% (synthetic cancer “breath” mixtures), 82% (lung cancer cell lines).	Ultrafast (<300 ms) neural decoding; high accuracy across complex VOC mixtures; surpasses many synthetic detectors	Mainly cell culture or synthetic data; invasive electrophysiology remains a challenge, along with stability and clinical translation.	[24,53,54]

Notes: Accuracy refers to correctly classified cases among all samples. Sensitivity = TP/(TP + FN), where TP = true positives and FN = false negatives. Specificity = TN/(TN + FP), where TN = true negatives and FP = false positives. “Overall accuracy” reflects the primary value reported, usually macro-averaged across classes. Cohort sizes (n) and sample sources are listed in each row. Validation type is specified: internal (same dataset) or external. Pilot- or company-reported studies are explicitly noted. For *C. elegans*, advanced pipelines (CNNs, LSTMs, RNNs) have been validated for worm tracking but not yet applied to cancer urine datasets, where logistic regression and threshold analyses remain standard. Because cancer screening involves class imbalance, AU-PRC is a more informative metric than accuracy; for example, the SpotitEarly canine AI–breath platform reported AU-PRC ≈ 0.96 (n ≈ 1400).

## Data Availability

No new data were created or analyzed in this study.

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
