# Peer review of "Beyond Binary: A Machine Learning Framework for Interpreting Organismal Behavior in Cancer Diagnostics"

_biomedicines, 2025, doi:10.3390/biomedicines13102409_

Round 1

Reviewer 1 Report

Comments and Suggestions for Authors

The paper is titled as “Beyond Binary: A Machine Learning Framework for Interpreting Organismal Behavior in Cancer Diagnostics” and it addresses the innovative concept of utilizing organismal biosensing and machine learning to improve cancer diagnostics. 

The topic is interesting and it integrates oncology, computational ethology, and artificial intelligence, representing one of the emerging research frontiers in precision oncology. And, it  fits well within the scope of Biomedicines JOurnal.

However, while the manuscript presents interesting perspectives and synthesizes interdisciplinary findings, I recommend Major Revision before reconsideration for publication. Specifically, the authors should: 

1) There is no clear problem/paper definition in the Introduction section.

2) The paper is constructed  as a conceptual essay than as a rigorously critical review. It should be structured as an academic paper.

3) One of the main expectations in a review article is a structured comparison of previous works. While the manuscript presents a wide range of studies on organismal biosensing and machine learning applications, it primarily describes them sequentially. The absence of systematic comparisons between studies is a major weakness.

4) The paper is titled as “Beyond Binary: A Machine Learning Framework for ...", therefore provide greater depth in the machine learning sections, with concrete case comparisons. the paper devotes large portions to biological background.

5) Table 1 is valuable, but several reported metrics lack context (e.g., sample sizes, validation schemes), making it difficult for readers to assess robustness.
Example Cancer Type--?
ML Method & Paradigm--  "Antennal-lobe neural recordings with supervised decoding" what is ML here?

6)
Figure 3. Machine learning paradigms for decoding organismal behavior. Pathway 2 of the dual-pathway framework integrates three paradigms: (a) feature-based learning....

There are no subfigures here as a,b,c   !

7) The Conclusion section of an academic paper aims to provide a concise summary of the study’s main findings, emphasize their significance within the broader field, and highlight the novel contributions made. It should also briefly acknowledge limitations and suggest directions for future research. In essence, the conclusion answers what was found, why it matters, and what should come next.
There is no Conclusion section in the paper.

8) Check the paper for language mistakes. Some of them are as follows:

Critically, these technological advance generated vast,... --Critically, these technological advances generated vast,
The resulting time series of body-part coordinates allows for the identification…--The resulting time series of body-part coordinates allow for the identification…
The primary argument of this review is that machine learning provides a framework  for decoding this rich behavioral signal.-- rich?
AI improves standardization and multi-cancer capability--?

Comments on the Quality of English Language

Check the paper for language mistakes. Some of them are as follows:

Critically, these technological advance generated vast,... --Critically, these technological advances generated vast,
The resulting time series of body-part coordinates allows for the identification…--The resulting time series of body-part coordinates allow for the identification…
The primary argument of this review is that machine learning provides a framework  for decoding this rich behavioral signal.-- rich?
AI improves standardization and multi-cancer capability--?

Author Response

Response to Reviewer 1

We are grateful to Reviewer 1 for recognizing the novelty of our manuscript and its interdisciplinary scope. In response to the detailed feedback, we have undertaken a major revision to improve structure, clarity, and depth.

Comment 1. “There is no clear problem/paper definition in the Introduction section.”
Response: We agree. The Introduction has been revised to provide a clear articulation of the central problem. In Section 1.3 (Problem Statement and the Limitation of Binary Readouts), we now explicitly outline three challenges that shaped the development of organismal biosensing: variability, rigor, and scalability. We also emphasize why early binary readouts, though essential for validation, are insufficient for precision oncology. This sets the stage for the paper’s central argument: the need to move beyond binary outputs toward richer machine learning enabled interpretations.

Comment 2. “The paper is constructed as a conceptual essay rather than as a rigorously critical review. It should be structured as an academic paper.”
Response: We agree with the reviewer’s observation. To address this, the manuscript has been fully restructured into a conventional academic review format with clearly defined sections. The revised outline is as follows:

  • Introduction (Section 1): Background, problem statement, and central thesis.
  • Methodology and Scope (Section 2): Criteria for inclusion and review boundaries.
  • Biological Foundations (Section 3): Comparative overview of canines, elegans, and insects.
  • Machine Learning Paradigms (Section 4): Organized into feature-based learning, representation learning, and unsupervised paradigms, with species-specific case studies.
  • Dual-Pathway Framework (Section 5): Translational model linking biology and computation.
  • Discussion (Section 6): Integration of cross-species insights, regulatory and ethical considerations, and future directions.
  • Conclusions (Section 7): Concise synthesis of findings, limitations, and next steps.

This restructuring ensures the manuscript follows the expectations of a critical academic review while maintaining clarity and logical flow across sections.

Comment 3. “One of the main expectations in a review article is a structured comparison of previous works. The manuscript primarily describes them sequentially. The absence of systematic comparisons between studies is a major weakness.”
Response: We fully agree with this important point. To address it, we reorganized the review to emphasize structured, side-by-side comparisons rather than sequential narrative. Specifically:

  • The original Table 1 has now been revised and expanded as Table 2 in the current version. It synthesizes the computational literature across organisms with row-level detail: ML method/paradigm, example cancer type, key reported metrics (with definitions), cohort size (n) and sample source, validation scheme (e.g., blinded case–control, cross-validation, external hold-out), as well as advantages, limitations, and references. This provides direct comparisons between machine learning approaches.
  • In addition, we have introduced a new Table 1, which organizes the biological and sensory features of nematodes, canines, and insects. Parameters include olfactory strategy, sensory complexity, data modality, throughput, scalability, key advantages/limitations, and the primary ML challenge for each platform.

To ensure transparent interpretation, we expanded the Notes section of Table 2 to:

  • Define accuracy, sensitivity, and specificity with explicit formulas (TP, TN, FP, FN).
  • Clarify that for elegans, advanced pipelines (CNNs, LSTMs, RNNs) are validated for worm tracking but not yet applied directly to cancer urine datasets, where logistic regression/threshold analyses remain standard.
  • Highlight that where class imbalance is present, AU-PRC is preferable to accuracy; for example, the SpotitEarly canine AI–breath platform reported AU-PRC ≈0.96 (n ≈ 1400).
  • In the main text (Sections 3.3 and 4.3), we now provide short comparative summaries that reference Table 1 (biology) and Table 2 (machine learning), making it easy for readers to navigate the comparative landscape.

These revisions transform what was previously a sequential description into a structured, systematic comparison, directly addressing the reviewer’s concern.

Comment 4. “The paper is titled as ‘Beyond Binary…’; therefore provide greater depth in the machine learning sections, with concrete case comparisons. The paper devotes large portions to biological background.”
Response: We agree with the reviewer’s concern and have substantially expanded and reorganized the machine learning sections to provide greater depth, balance, and concreteness. Specifically:

  • Section 4 (Machine Learning Paradigms) has been restructured into three subsections:
    • 2.1 Feature-based supervised learning: highlights interpretable pipelines (e.g., logistic regression, SVMs, random forests) with examples from nematode chemotaxis and canine alert/no-alert classification. We emphasize how attribution methods (SHAP, LIME) map predictions to biologically meaningful features.
    • 2.2 Representation learning: details deep neural networks (CNNs, RNNs, LSTMs, Transformers) applied to high-dimensional behavioral and electrophysiological data, with case comparisons between species.
    • 2.3 Unsupervised and hybrid learning: explains clustering, latent embeddings, and hybrid semi-supervised methods, and their roles in discovery and generalization.
  • Section 4.3 (Case studies) provides organism-specific comparisons:
    • elegans: chemotaxis indices analyzed with logistic regression and calcium imaging classified with SVMs; pilot pipelines with CNNs and RNNs for behavioral tracking.
    • Canines: ANN-trained GC–MS data and AI-assisted ensemble models in breath screening.
    • Insects: antennal-lobe neural recordings decoded with supervised classifiers (SVMs, Random Forests).
      Each subsection cites cohort size, validation design, and reported performance metrics, ensuring transparency.
  • Table 2 (previously Table 1) now directly supports this expansion by summarizing machine learning applications across species, with explicit n, cancer type, validation scheme, metrics, advantages, and limitations. The Notes section clarifies metric definitions, highlights AU-PRC, and specifies where pipelines are validated for tracking but not yet applied to cancer urine datasets.

Together, these revisions provide the greater depth in machine learning content and concrete case comparisons that the reviewer requested, while balancing the biological background with computational rigor.

Comment 5. “Table 1 is valuable, but several reported metrics lack context (e.g., sample sizes, validation schemes).”
Response: We thank the reviewer for this observation. The original Table 1 is now revised and presented as Table 2 (Section 4.3). Key improvements include:

  • Cohort sizes (n): Now explicitly added in each row (e.g., n = 242; n = 78; n ≈ 1400; n ≈ 200; n ≈ 20–30).
  • Validation schemes: Case–control vs. prospective study designs, and internal vs. external validation, are clearly stated (external validation marked with *).
  • Metric definitions: Sensitivity, specificity, and accuracy are standardized and defined in the expanded Notes line beneath Table 2.
  • Advanced ML clarification: For elegans, CNN/LSTM/RNN pipelines are noted as validated for worm tracking but not yet applied to cancer urine datasets.
  • Pilot/company-reported studies: Explicitly flagged in the Notes to distinguish feasibility or non–peer-reviewed sources.
  • AU-PRC: The Notes now emphasize AU-PRC as a more informative metric under class imbalance, with an example provided (SpotitEarly canine AI–breath platform, AU-PRC ≈ 0.96, n ≈ 1400).

These changes improve transparency and comparability, allowing readers to assess robustness at a glance.

Comment 6. “Figure 3. Machine learning paradigms for decoding organismal behavior. Pathway 2 of the dual-pathway framework integrates three paradigms: (a) feature-based learning…. There are no subfigures here as a, b, c!”

Response: Thank you for pointing this out. Figure 3 has been revised to include clearly labeled subfigures:

  • (a) Feature-based supervised learning
  • (b) Representation learning with deep neural networks
  • (c) Unsupervised discovery methods

The figure caption has also been updated to explicitly reference these subfigures, ensuring alignment between text and figure labeling. This revision clarifies the visual structure of the framework and directly addresses the reviewer’s concern.

Comment 7. “The Conclusion section of an academic paper aims to provide a concise summary of the study’s main findings, emphasize their significance within the broader field, and highlight the novel contributions made. It should also briefly acknowledge limitations and suggest directions for future research. There is no Conclusion section in the paper.”

Response: We agree with this important point. A dedicated Section 7 (Conclusions) has been added to the revised manuscript. This section:

  • Summarizes key findings: integration of organismal biosensing with machine learning, identification of the bottleneck of binary readouts, and the potential of high-dimensional behavioral decoding.
  • Highlights significance: positions this interdisciplinary approach within the broader context of precision oncology and biosensing innovation.
  • Acknowledges limitations: small sample sizes, inter-laboratory variability, class imbalance, and limited external validation across models.
  • Outlines future directions: multimodal data integration, synthetic biology approaches, AI-assisted phenotyping, and development of lightweight ML pipelines for clinical translation.

The addition of Section 7 provides the structured and forward-looking conclusion expected in an academic review article.

Comment 8. “Check the paper for language mistakes. Some of them are as follows: ‘technological advance’ → ‘technological advances’; ‘coordinates allows’ → ‘coordinates allow’; ‘rich behavioral signal’—rich?; ‘AI improves standardization and multi-cancer capability—?’”

Response: We carefully revised the manuscript for grammar, clarity, and tone. All specific examples flagged by the reviewer have been corrected

Additionally, we conducted a full proofreading pass to eliminate other grammatical errors and ambiguous phrasing. This has improved readability and alignment with the formal style of Biomedicines.

Additional clarification requested. “Antennal-lobe neural recordings with supervised decoding — what is the ML method here?”

Response: We clarified this point in the revised manuscript. In Section 4.3.3 (Insects: A Case Study in Neural Decoding) and in Table 2 (Summary of machine learning applications in organismal biosensing), we now specify that antennal-lobe recordings were decoded using Support Vector Machines (SVMs) and Random Forest classifiers, which consistently outperformed rate-based metrics in VOC discrimination tasks.

Reviewer 2 Report

Comments and Suggestions for Authors

The manuscript is timely and clearly written; the suggestions aim to strengthen transparency, reproducibility, and readability without altering the core message.

First, the abstract and Table 1 would benefit from tighter quantitative context. The abstract currently highlights headline figures (e.g., ~97% for nematode calcium imaging, ~94% for AI–canine breath screening, and millisecond-scale insect classification) but does not pair them with cohort sizes, definitions of the metrics used (accuracy vs. sensitivity/specificity/AUC), confidence intervals, or whether any external validation was performed. Please add (i) n and cohort/source descriptors next to each abstract statistic, and (ii) a concise table-footnote schema that defines every metric used and flags any external test-set results; consider including AU-PRC where class imbalance is likely. Table 1 already reports canine “overall accuracy” and per-modality sensitivity/specificity and includes a notes line—expanding that notes line to standardize definitions across rows would materially help readers interpret results at a glance.

Second, the “behavioral vector” concept would land more smoothly if a one-sentence operational definition appeared at the first mention (end of §1 or in the caption of the introductory figure), not only later in §3.2. Right now, the abstract cues a high-dimensional “behavioral vector,” while the formal state-space definition of B(t)B(t) arrives much later; pre-placing a compact definition (e.g., “a time-indexed vector whose dimensions summarize posture/orientation/kinematics with temporal structure”) plus one concrete feature example (reorientation frequency, rapid turns, etc.) would help non-specialist readers track the argument until §3.2.

Third, because the paper argues persuasively for standardization, a small, actionable checklist in the Supplement (one page) would greatly increase practical utility. Consider a “ready-to-run” grid that lists: sample handling (e.g., urine/breath storage at 4 °C ≤24 h or immediate freezing), environmental controls (temperature/humidity), organism handling/training (noting reported 10–15% swings in canine detection accuracy with training conditions), and assay-specific parameters (e.g., C. elegans agar concentration 2% vs 2.5%, odor-gradient stability, plate preparation). You already motivate microfluidic arenas and controlled scent-presentation chambers as mitigation; place explicit model recommendations and minimal specs in that checklist, with links to community resources.

Fourth, please fine-tune conflict-of-interest (COI) and industry-update citations to preserve a neutral review tone. The COI statement correctly notes multiple authors’ employment at Hirotsu Bio Science (developer of N-NOSE). In the case-study section, one reference is an “Innovation Update”/press notice; it is suitable to mention but should be explicitly labeled as non–peer-reviewed, with any performance claims summarized cautiously in main text and fuller details (URL, slide decks) moved to the Supplement. A single sentence flagging the evidence tier will safeguard reader trust.

Fifth, a few editorial touch-ups will polish the presentation. In §5’s opening, tighten the repetition (“propose a Dual-Pathway Framework … apply the framework proposed here”). In §1.3, change “technological advance” to “technological advances.” In Table 1, extend the notes line to define “overall accuracy” (macro vs. micro averaging; per-sample vs. per-class) and point readers to alternative metrics for imbalance (AU-PRC). 

Author Response

We thank Reviewer 2 for the detailed and practical suggestions, which have significantly improved the clarity, consistency, and utility of the manuscript.

Comment 1. “Abstract and Table 1 would benefit from tighter quantitative context.”

Response: The Abstract has been revised so that each reported performance figure is paired with explicit sample size, study design, and validation scheme. For example:

  • “C. elegans urine chemotaxis: sensitivity 95.8%, specificity 95.0% (n = 242; blinded case–control).”

In addition, the comparative tables have been expanded to ensure quantitative transparency:

  • Table 1 (Comparative features of biosensing platforms) now provides structured cross-species comparison.
  • Table 2 (Summary of machine learning applications; formerly Table 1 in the original submission) now reports sample sizes (n), sensitivity/specificity values, validation types (internal vs. external, with * denoting external), and includes an expanded Notes section clarifying metric definitions, study design, and external validation.

These revisions make the statistical context of all reported metrics more transparent and reproducible.

Comment 2. “The ‘behavioral vector’ concept should be defined earlier.”

Response: We thank the reviewer for this suggestion. To improve clarity, we now introduce the concept at its first mention in Section 1.4, where it is defined as:

“The behavioral vector, B(t), is a time-indexed, multidimensional representation of organismal behavior, where each dimension corresponds to a quantifiable feature (e.g., posture, orientation, trajectory), and temporal dynamics (e.g., reversal frequency, pirouettes) provide additional discriminatory power.”

The notation B(t) is therefore established early in the manuscript. In addition, the caption of Figure 1 has been revised to include a one-sentence recap for immediate context:

“Figure 1. Conceptual overview of organismal biosensing. The behavioral vector, B(t), represents a time-indexed, multidimensional description of posture, orientation, and kinematic features, providing the input for machine learning models.”

Section 4.1 then expands on this with the full state-space treatment.

Comment 3. “Provide a small, actionable checklist in the Supplement.”

Response: We appreciate this suggestion and have created Supplementary Table S1, a one-page reproducibility checklist designed for practical use. It covers four domains—sample handling, environmental controls, organism training/handling, and assay-specific parameters. Each entry specifies recommended specifications, accompanying notes, and references.

This table is intended as a ready-to-run grid that can be adopted directly in experimental protocols, thereby aligning with the reviewer’s request for actionable guidance and enhancing reproducibility across laboratories.

Comment 4. “Fine-tune COI and industry citations.”
Response: Section 4.3.2 now explicitly labels the Hirotsu Bio Science “Innovation Update” as non–peer-reviewed industry material. Claims are summarized cautiously in main text, with details moved to Supplementary Table S2. This preserves neutrality and transparency.

Comment 4. “Fine-tune COI and industry citations.”

Response: We thank the reviewer for this important point. Section 4.3.1 now explicitly labels the Hirotsu Bio Science “Innovation Update” as non–peer-reviewed industry material. The revised text reads:

“Preliminary data from Hirotsu Bio Science (September 2024 ‘Innovation Update’) are cited here as non–peer-reviewed industry material. These claims are summarized cautiously, with full details (URLs and supporting slides) provided in the Supplementary Notes.”

This ensures that performance claims are clearly distinguished from peer-reviewed evidence. In addition, we reviewed the Conflict of Interest statement, which already discloses all relevant author affiliations and remains unchanged. Together, these revisions safeguard neutrality and preserve transparency regarding the evidence tier.

Comment 5. “A few editorial touch-ups will polish the presentation.”

Response: We thank the reviewer for these suggestions. All edits have been made. The opening of Section 5 has been streamlined to remove redundancy; the revised text now reads:

“We propose a Dual-Pathway Framework as a practical strategy for clinical implementation, integrating both validated low-dimensional screening metrics (Pathway 1) and AI-augmented high-dimensional analyses (Pathway 2).”

In addition, Section 1.3 now uses the corrected phrasing “technological advances”, and the notes line of Table 1 has been expanded to define “overall accuracy” (macro vs. micro averaging; per-sample vs. per-class) and to highlight AU-PRC as a more informative metric in imbalanced datasets.

Round 2

Reviewer 1 Report

Comments and Suggestions for Authors

The authors made related corrections.
It can be accepted as is.